# High Expression of miR-34a Associated with Less Aggressive Cancer Biology but Not with Survival in Breast Cancer

**DOI:** 10.3390/ijms21093045

**Published:** 2020-04-26

**Authors:** Yoshihisa Tokumaru, Eriko Katsuta, Masanori Oshi, Judith C. Sporn, Li Yan, Lan Le, Nobuhisa Matsuhashi, Manabu Futamura, Yukihiro Akao, Kazuhiro Yoshida, Kazuaki Takabe

**Affiliations:** 1Breast Surgery, Department of Surgical Oncology, Roswell Park Comprehensive Cancer Center, Buffalo, NY 14263, USA; yoshitoku1090@gmail.com (Y.T.); eriko.katsuta@roswellpark.org (E.K.); masanori.oshi@roswellpark.org (M.O.); judith.sporn@roswellpark.org (J.C.S.); lanle@buffalo.edu (L.L.); 2Department of Surgical Oncology, Graduate School of Medicine, Gifu University, Gifu 501-1194, Japan; nobuhisa517@hotmail.com (N.M.); mfutamur@gifu-u.ac.jp (M.F.); kyoshida@gifu-u.ac.jp (K.Y.); 3Department of Gastroenterological Surgery, Yokohama City University Graduate School of Medicine, Yokohama 236-004, Japan; 4Department of Biostatistics & Bioinformatics, Roswell Park Comprehensive Cancer Center, Buffalo, NY 14263, USA; li.yan@roswellpark.org; 5Department of Surgery, University at Buffalo Jacobs School of Medicine and Biomedical Sciences, The State University of New York, Buffalo, NY 14203, USA; 6United Graduate School of Drug and Medical Information Sciences, Gifu University, Gifu 501-1194, Japan; yakao@gifu-u.ac.jp; 7Department of Surgery, Niigata University Graduate School of Medical and Dental Sciences, Niigata 951-8510, Japan; 8Department of Breast Surgery and Oncology, Tokyo Medical University, Tokyo 160-8402, Japan; 9Department of Breast Surgery, Fukushima Medical University School of Medicine, Fukushima 960-1295, Japan

**Keywords:** microRNA-34a, miR-34, METABRIC, apoptosis, p53, EMT, Cell cycle, GSEA, GSVA, TCGA

## Abstract

Most breast cancer (BC) patients succumb to metastatic disease. MiR-34a is a well-known tumor suppressive microRNA which exerts its anti-cancer functions by playing a role in p53, apoptosis induction, and epithelial-mesenchymal transition (EMT) suppression. Molecular Taxonomy of Breast Cancer International Consortium (METABRIC) and The Cancer Genome Atlas (TCGA) cohorts were used to test our hypothesis that miR-34a high BCs translate to less aggressive cancer biology and better survival in large cohorts. There was no association between miR-34a expression levels and clinicopathological features of BC patients except for HER2 positivity. MiR-34a high expressing tumors were associated with lower Nottingham pathological grades and lower MKI67 expression. In agreement, high miR-34a tumors demonstrated lower GSVA scores of cell cycle and cell proliferation-related gene sets. High miR-34a tumors enriched the p53 pathway and apoptosis gene sets. Unexpectedly, high miR-34a tumors also associated with elevated EMT pathway score and ZEB1 and two expressions. MiR-34a expression did not associate with any distant metastasis. Further, high miR-34a tumors did not associate with better survival compared with miR-34a low tumors. In conclusion, the clinical relevance of miR-34a high expressing tumors was associated with suppressed cell proliferation, enhanced p53 pathway and apoptosis, but enhanced EMT and these findings did not reflect better survival outcomes in large BC patient cohorts.

## 1. Introduction

Breast cancer is the second leading cause of cancer-related deaths among women in the United States [1]. Even though the five-year survival rate of breast cancer patients is over 90%, approximately 40,000 patients succumb to this disease annually in the United States [1,2]. From the fact that most breast cancer patients succumb to metastatic or recurrent disease, inhibition of metastasis is a key strategy to decrease breast cancer-related deaths [3]. Given the complexity of pathobiology of metastasis that greatly differs between subtypes or even on an individual basis, it is essential to clarify what molecules participate in the formation of cancer metastasis and patient survival.

MicroRNAs (miRNAs) are a type of non-coding RNA that regulates the gene expression of target genes at the post-transcriptional level [4,5]. MiRNAs silences the gene expression by binding to 3′-untranslated regions of messenger RNA (mRNA) or inducing the degradation of mRNA. Tumor suppressive miRNAs are reported to be dysregulated in various cancers [4]. Our group has been reporting the clinical relevance of the expression of tumor-suppressive miRNAs using computational biological approaches [6,7,8,9,10].

MiR-34a is one of the tumor-suppressive miRNAs that exerts its anti-cancer function by playing a role in a well-known tumor suppressor, p53 [11]. Preclinical in vitro studies demonstrated that p53 regulates miR-34a expression [12] and miR-34a contributes to p53-mediated apoptosis [12,13]. miR-34a was also demonstrated to regulate cell cycle and suppress cell proliferation [14]. Further, previous studies demonstrated that miR-34a inhibits epithelial-mesenchymal transition (EMT), the mechanism of how the cancer cells develop distant metastases, by suppressing EMT-related transcriptional factors, such as TWIST1 and ZEB 1 [15,16]. Although the number of preclinical in vitro studies demonstrated that overexpression of miR-34a in breast cancer leads to suppression of cell proliferation, induction of apoptosis, inhibition of EMT as mentioned above, the clinical relevance of miR34a high expressing tumor remains ambiguous. 

In this study, we hypothesized that miR-34a functions as tumor-suppressive miRNA in human breast cancer cohorts and associate with less cell proliferation that leads to favorable survival outcome. 

## 2. Results

### 2.1. Clinicopathological Features Were Not Significantly Different between miR-34a High and miR-34a Low Expressing Tumors Except for HER2 Expression

The association between miR-34a expression levels and clinicopathological features of breast cancer patients in the Molecular Taxonomy of Breast Cancer International Consortium (METABRIC) cohort was analyzed. Patients were divided into two groups using a median cutoff of miR-34a expression. Although miR-34a has been reported to function as a tumor-suppressive microRNA, its expression levels did not associate with the age, tumor size, lymph node metastasis, or cancer staging (Table 1). HER2 receptor positivity alone demonstrated statistical difference by miR-34a expression levels among subtypes. In The Cancer Genome Atlas (TCGA) cohort, older patients were significantly associated with low miR-34a expression, and Luminal A was significantly associated with low whereas unknown was associated with high miR-34a in PAM 50 classification, which may contribute to a potential bias (Appendix A).

### 2.2. Nottingham Pathological Grade Was Associated with miR-34a Expression and miR-34a High Expressing Tumors Were Associated with Lower Expression of MKI67 in METABRIC

Despite high levels of miR-34a not associating with the clinical aggressive parameters, we hypothesized that tumors with high expression of miR-34a associate with pathological and molecular markers of cancer cell proliferation given that miR-34a is a tumor-suppressive miRNA. Nottingham pathological grade represents the histological aggressiveness of a bulk tumor. As we have expected, lower-grade tumors were associated with higher miR-34a expression in the whole METABRIC cohort (*p* < 0.01; Figure 1A). Among the subtypes, estrogen receptor (ER)-positive/HER2−negative (ER+/HER2−) and triple-negative (TN) demonstrated significant association of miR-34a high expression with low grade tumors (*p* < 0.05, and *p* < 0.01, respectively), whereas HER2-positive/ER-negative (HER2+/ER−) did not (*p* = 0.314). MKI67 gene codes ki-67, which is one of the most commonly used molecular markers of cancer cell proliferation. High miR-34a expression tumors significantly associated with lower expression of MKI67, which indicates lower cell proliferation in the whole cohort (*p* < 0.001, Figure 1B). Interestingly, miR-34a high expression tumors were associated with lower MKI67 expression levels in all the subtypes. Statistical significance was not reached in either grade or MKi67 other than TN in the TCGA cohort, which possesses only half the number of patients of the METABRIC cohort (Appendix A). The tumors with high miR-34a expression have less cell proliferation in METABRIC.

### 2.3. High MiR-34a Breast Cancer Associated with Lower Gene Set Variation Analysis (GSVA) Scores of the Gene Sets Related with Cell Cycle or Cell Proliferation

Based on the above result that high miR-34a breast cancer associated with lower cell proliferation, we hypothesized that cell cycle and cell proliferation are suppressed in those tumors. Gene sets variation analysis (GSVA) allows us to not only explore the underlying mechanisms but also to generate a score to estimate and quantify the amount of involvement of the pathway [17]. High miR-34a expression tumors associated with significantly lower scores of E2F_TARGETS, G2M_CHECKPOINT, and MITOITC_SPIMDLE, all of which are gene sets related with cell cycle or cell proliferation, with the whole cohort (*p* < 0.001, *p* < 0.001, and *p* < 0.001, respectively; Figure 2). Interestingly, ER+/HER2− and TN, but not HER2+/ER− subtype demonstrated the significant association of miR-34a high expression tumors and lower GSVA scores, which were the subtypes that demonstrated a statistically significant association with both grade and MKI-67 expression (Figure 2). These results support the notion of miR-34a as a tumor-suppressive microRNA and its high expression tumors demonstrate lower cell proliferative features. However, statistical significance was not reached except for TN in the TCGA cohort that has only half the sample size compared with METABRIC (Appendix A). 

### 2.4. High MiR-34a Expression Tumors Enriched p53 Pathway and Apoptosis Gene Sets and Was Associated with Expression of Apoptosis Related Genes 

It has been reported that p53 regulates the expression levels of miR-34a [18]. Given that p53 is mutated in various cancers, it was of interest whether p53 mutation status has any effect on miR-34a expression in human breast cancer. We found that there was no association between miR-34a expression levels and mutation status of p53 in whole or in any of the subtypes in the METABRIC cohort, which is in agreement with the previous report [19] (Figure 3A). It was reported that overexpression of miR-34a activated the p53 pathway in preclinical in vitro system [12]. We hypothesized that this is also the case in patients and the p53 pathway is activated in high miR-34a expression tumors in METABRIC and TCGA cohorts. As expected, GSEA gene sets related to the p53 pathway were significantly enriched with miR-34a high expression tumors compared to miR-34a low expression tumors in both METABRIC and TCGA cohorts (Figure 3B). Since p53 mediates apoptosis, we estimated the apoptosis by utilizing the GSVA score of APOPTOSIS and expression of apoptosis-related genes [20]. High-miR-34a expression tumors demonstrated higher APOPTOSIS score in the METABRIC cohort, but not in the TCGA cohort, may be due to a smaller sample size (*p* < 0.001 and *p* = 0.121, respectively; Figure 3C). We further examined the relationship between apoptosis and miR-34a expression by analyzing the expression of apoptosis-related genes, such as Bcl-2, Annexin 5, Caspase 8, and Caspase 9 using the METABRIC cohort. The expression levels of the anti-apoptotic gene, Bcl-2, were significantly lower in miR-34a high expression tumors (*p* < 0.05; Figure 4). On the contrary, the expression levels of apoptotic genes, Annexin 5, Caspase 8, and Caspase 9 were upregulated in miR-34a high expression tumors (*p* < 0.05, *p* < 0.001, and *p* < 0.001, respectively; Figure 4). Among the subtypes, miR-34a associated with elevation of ANXA5 and CASP9 in ER+/HER2−, with elevation of CASP9 alone in HER2+/ER−, and with suppression of BCL2 in TN (Figure 4). Overall no difference was seen between the high and low expression of miR-34a in apoptosis-related gene expressions in smaller size TCGA cohort (Appendix A). 

### 2.5. High MiR-34a Expression Tumors Associated with Higher Epithelial–Mesenchymal Transition (EMT) Score

Previous preclinical studies have demonstrated that overexpression of miR-34a inhibited metastasis by suppression of EMT relating factors, such as Notch1, TWIST1, and ZEB1. We hypothesized that miR-34a high tumors are associated with lower expression of EMT relating factors. As we have expected, miR-34a high expression tumors demonstrated lower expression levels of EMT inducing factor, Notch-1, in whole METABRIC cohorts, as well as in all the subtypes of breast cancer patients (*p* < 0.001, *p* < 0.001, *p* < 0.05, and *p* < 0.01, respectively; Figure 5). However, to our surprise, high miR-34a expression tumors were associated with higher expression of the other EMT relating genes, such as ZEB1 and ZEB2 in the whole cohort as well as ER+/HER2− and TN subtypes. MiR-34a high tumors did not exhibit any consistency with TWIST1. 

We further examined the trend by utilizing the GSVA score of the EMT gene set. Interestingly, miR-34a high tumors were associated with a higher score of the EMT gene sets with the whole cohort as well as ER+/HER2− and TN subtypes. These results imply that overall miR-34a high expression tumors associate more with EMT pathway promoting gene expressions than miR-34a low expression tumors. Statistically significant difference was not achieved in a smaller TCGA cohort except Notch1 in TN subtype (Appendix A). We further analyzed the association between miR-34a and EMT pathway. In the METABRIC cohort, high miR-34a expression associated with elevated EMT only in Stage I-III, but not Stage IV, which suggests that this association does not occur in the tumor with metastasis at the time of diagnosis (Appendix A). On the other hand, high miR-34a associated with enhanced EMT pathway in both patients that did not develop any recurrence/metastasis within five years and that did, which suggests that this association occurs regardless of whether the patients develop recurrence/metastasis or not. This relationship between miR-34a expression and EMT pathway was not significant in a smaller TCGA cohort regardless of whether there was metastasis at diagnosis or not, nor whether the patients developed metastasis or not (Appendix A). It is known that EMT is associated with Claudin-low subtype. Thus, analysis of the relationship between miR-34a expression and EMT pathway in Claudin-low vs. other subtypes would be of interest. Unfortunately, we were unable to conduct this analysis since Claudin-low status was not included in parameters available in METABRIC or TCGA. 

### 2.6. High miR-34a Expression Tumor Was Not Associated with Distant Metastasis and Was Not Associated with Better Survival Outcome

Given that high miR-34a expression tumor associated with enhancement of the EMT pathway gene set, we assessed the clinical relevance of miR-34a expression by patient survival. Disease-free survival (DFS) of specific distant metastasis sites, such as bone and lung and other metastasis sites were analyzed in the TCGA cohort. MiR-34a high expression tumors did not associate with less metastasis in any of the sites (*p* < 0.253, *p* < 0.763, and *p* < 0.263, respectively; Figure 6A). High miR-34a expression tumor did not associate with better DFS including TN in both TCGA and METABRIC cohorts. HER2+/ER− subtype in the TCGA cohort alone demonstrated an association of high miR-34a tumors and better DFS (*p* = 0.036; Figure 6B), however, this result was not validated with METBRIC cohort (*p* = 0.057; Figure 6B) which possesses larger sample size than TCGA. There was no association of miR-34a high expressing tumors with better OS compared with miR-34a low expressing tumors, except it associated with worse OS in ER+/HER2− subtype in TCGA cohort (*p* = 0.025; Figure 6B). This result was not consistent in METABTRIC (*p* = 0.123; Figure 6B). We found that high miR-34 expression tumor has significantly more patients older than 65 years old in the TCGA cohort (Appendix A), but not in METABRIC (Table 1). 

## 3. Discussion 

MiR-34a is a well-known tumor-suppressive miRNA. There were no clinicopathological features except HER2 status associated with miR-34a expression levels. MiR-34a high expression tumors were associated with lower Nottingham pathological grades and lower MKI67 expression, which are markers of cell proliferation. These results were echoed by miR-34a high expression tumors associated with lower GSVA scores of the gene sets relating cell cycle and cell proliferation, such as E2F_TARGETS, G2M_CHECKPOINT, and MITOTIC SPINDLE. High miR-34a expression tumors enriched the p53 pathway and apoptosis gene sets and associated with expressions of apoptosis-related genes, which agree with previous reports. On the other hand, miR-34a high expression tumors associated with an enhanced EMT pathway, which was the opposite of the previous reports. MiR-34a expression did not associate with bone, lung, or other distant metastases in the TCGA cohort. High miR-34a expressing tumor did not associate with better survival compared with miR-34a low expressing tumors.

In HER2 overexpressing subtype, miR34a high expressing tumors were significantly associated with Ki67 but not with cell progression pathways in GSVA. This may be because HER2 positivity was significantly high among miR-34a high group, which is a confounding factor to begin with. Since HER2 overexpressing subtype is well known to be biologically aggressive and miR-34a is known to be a suppressive-miR, it is not surprising to imagine that these two forces in opposite directions may have counterbalanced and resulted in no difference in cell progression pathways in GSVA. Even for MKi67, it was barely significant (*p* = 0.032) whereas the other subtypes were more significant (*p* < 0.001). Further, MKi-67 represents only one gene although it is used as a cell proliferative marker in the clinics, whereas the GSVA score demonstrates the result of gene set which includes hundreds of genes associate with cell proliferation. To this end, we believe that GSVA reflects the broader biological activity of cell proliferation compared with MKi-67 gene expression alone. Therefore, we concluded that there was no association between high miR-34a expressing tumors and cell proliferation with HER2+/ER− subtype.

Previously, our group has assessed and reported clinical relevance of tumor-suppressive miRNAs, such as miR-9, miR-30a, miR-200c, miR-18a, miR-205, and miR-744 in breast cancer by utilizing computational biological approaches [6,7,8]. Accumulating data from preclinical in vitro studies have shown that tumor-suppressive miRNAs play a critical role in breast cancer progression and metastasis. However, only a few studies are conducted to assess the clinical relevance of those miRNAs. We believe that translational studies to validate miRNA functions discovered in the in vitro setting, especially in large sample size human patient cohorts, is essential to utilize the basic science results in clinical practice, and the computational biological approach is one of the powerful tools that are available to us at this time. 

Peurala et al. demonstrated that high miR-34a expression was associated with ER−negativity, HER2−positivity, positive nodal status, and high tumor grade utilizing a cohort containing a total of 1172 breast cancer patients [21]. Even though HER2−positivity was consistent with the present study, we did not observe an association with ER−negativity nor positive nodal status. MiR-34a high expressing tumors were associated with low Nottingham pathological grade in our analysis, which was the opposite of the previous study. One of the possible reasons for these differences may be due to the detection method of miR-34a expression. The current study used microRNA-sequence data of bulk tumors, whereas their report used locked nucleic acid in situ hybridization that utilizes fixed paraffin-embedded breast cancer tissue.

Previous preclinical in vitro studies demonstrated that p53 regulates miR-34a expression [12] and miR-34a induces p53-mediated apoptosis [13,19]. In concordance with this, miR-34a high-expressing tumors enriched p53-related gene sets which implies that the p53 pathway is activated with miR-34a high expression tumors. Given p53 is a well-documented proapoptotic factor, we assessed the association of miR-34a tumors and apoptosis using clinical samples. MiR-34a high expression tumors associated with a higher score of APOPTOSIS gene set with the METBRIC cohort which implies of promotion of apoptosis. Analysis of apoptosis-related gene expressions followed the same trend despite the difference was very small for some genes. Association of miR-34a high tumors and high apoptosis score was not reproduced by the TCGA cohort, where its sample is half the size of METABRIC. 

EMT is a critical mechanism in the metastasis of cancer [22]. Previous preclinical in vitro studies demonstrated that miR-34a directly targets Notch1, negatively regulates its expression and leads to inhibition of EMT [22,23]. This notion was echoed with clinical patient data analysis of our current study in which miR-34a high expressing tumors demonstrated lower expression levels of NOTCH1 compared with miR-34a low expressing tumors. On the contrary, miR-34a high expressing tumors demonstrated higher expression of ZEB1 and ZEB2 genes, another EMT related genes, as well as enhanced EMT in the current study, despite miR-34a inhibited EMT by suppressing TWIST1, ZEB family in the previous study [16,24]. This discrepancy may be due to the fact that preclinical studies analyze protein expression by western blotting whereas the current study analyzes a transcriptome expression of a bulk tumor using the computational biological method. The benefit of this bioinformatical approach using transcriptome is that it enables us to analyze the whole cohort of thousands of samples instead of analyzing a couple of cell-lines. Further, since this method utilizes mathematical calculations, there is less possibility of human technical error on the assay. On the other hand, its limitation is that our analyses are on transcriptomes and not protein, which is the final product that regulates cell biology.

Several studies reported the survival benefit of miR-34a high expressing tumors with triple-negative breast cancers [25,26,27,28]. Surprisingly this was not reproduced in the current study even though both present and previous studies used the median cutoff of miR-34a expression to determine miR-34a high expressing tumor. This discrepancy may be explained by the difference in sample size. The present study possessed the highest number of patients to analyze the survival benefit of miR-34a high expression by utilizing the METBRIC cohort which includes 207 triple-negative breast cancer patients’ survival data, whereas the largest cohort from the previous report was 109 cases [28]. 

When we had access to only a single small cohort (TCGA) with microRNA expression data, that was all we were able to publish at that time [7,8,9]. Clearly, for a study that developed a score using a testing cohort, that score needs to be validated with another cohort as we have published previously [10]. On the other hand, the current study is simply comparing high and low levels of miR-34a expression, thus there is no testing cohort to develop a score, nor validation cohort. With that said, there is no doubt that the results are stronger and convincing when they are consistent among multiple cohorts. We have analyzed METABRIC and TCGA, the two largest breast cancer cohorts with microRNA expression data to our knowledge. As described in result sections, not all the data were consistent. This may be partly due to the fact that METABRIC has gene expression microarray data that includes only miR-34a-3p, whereas TCGA has microRNA-sequence data that includes both miR-34-3p and -5p, or because the sample size of TCGA is half of that of METABRIC. High miR-34a expression was associated with age and PAM50 classification only in TCGA, which may have contributed to the disagreement.

There are some limitations to the current study. This study is a retrospective study which utilizes the publicly available database such as METABRIC and TCGA. We utilized the transcriptome expression data to assess the association of miR-34a and genes of interest, however previous reports use the protein expression to explore that association. There may be a discrepancy between expression between mRNA and protein level of genes of interest. The addition of protein expression analyses to verify some of the gene sets would strengthen our data, however, this was not possible due to lack of access to tissue samples.

In conclusion, the clinical relevance of miR-34a high expression tumors was associated with lower cell proliferation, enhanced p53 pathway and apoptosis, as well as EMT, however, these findings did not contribute to better survival outcomes.

## 4. Material and Methods

### 4.1. Obtaining Data of TCGA and METABRIC

We identified a total of 1282 patients from the Molecular Taxonomy of Breast Cancer International Consortium (METABRIC) database that have microRNA expression data as previously described [10,29,30]. From The Cancer Genome Atlas (TCGA), 755 patients were identified to have microRNA expression data and were extracted through cBioPortal and Broad Institute Firehose (http://gdac.broadinstitute.org/) as described previously as previously described [31,32,33,34,35]. The receptor status was obtained from the clinical parameters included in METABRIC and TCGA, which were determined by standard immunohistochemistry (IHC) and pathological analysis as we previously published [6,9,10,17,29,30,33,34,35,36,37]. Its correspondence with PAM50 is included in Table 1 and Appendix A. In the METABRIC cohort, ER-positive/HER2 negative (ER+/HER2−), HER2-positive/ER-negative (HER2+/ER−), and triple-negative (TN) subtypes had 907, 86, and 207 patients, respectively. In the TCGA cohort, ER+/HER2−, HER2+/ER−, and TN subtypes had 407, 30, and 117 patients, respectively. The median follow-up of METABRIC and TCGA, which are the medians of survival, was 188.07 and 26.02 months, respectively.

### 4.2. Gene Set Variation Analysis (GSVA) and Gene Set Enrichment Analysis (GSEA)

Gene Set Variation Analysis (GSVA) is a method to estimate the variation of gene set enrichment through the samples of expression data set [17,36]. GSVA enables us to understand the biological activity of the gene set of interest [17]. GSVA scoring of Hallmark gene sets, such as E2F_TARGETS, G2M_CHECKPOINT, MITOTIC_SPINDLE, APOPTOSIS, and EPITHELIAL_MESENCHYMAL_ TRANSITION was utilized in this study (Figure 2, Figure 3C, Figure 5, and Appendix A)

Broad Institute (http://software.broadinstitute.org/gsea/index.jsp) provides the publicly available software to perform Gene set enrichment analysis (GSEA). We performed GSEA using this publicly available software as previously described [6,9,35,37,38,39]. We used the p53 pathway of the Hallmark gene set in this study (Figure 3B).

### 4.3. Statistical Analysis

All the statistical analysis was performed using publicly available software, R software (http:///www.r-project.org/). For the survival analysis, we performed Kaplan-Meier survival analysis using greyzoneSurv packages in R. Fisher’s exact test was used for the analysis between the two groups, miR-34a high expressing tumors and miR-34a low expressing tumors. A two-sided *p*-value < 0.05 was considered statistically significant.

## Figures and Tables

**Figure 1 ijms-21-03045-f001:**
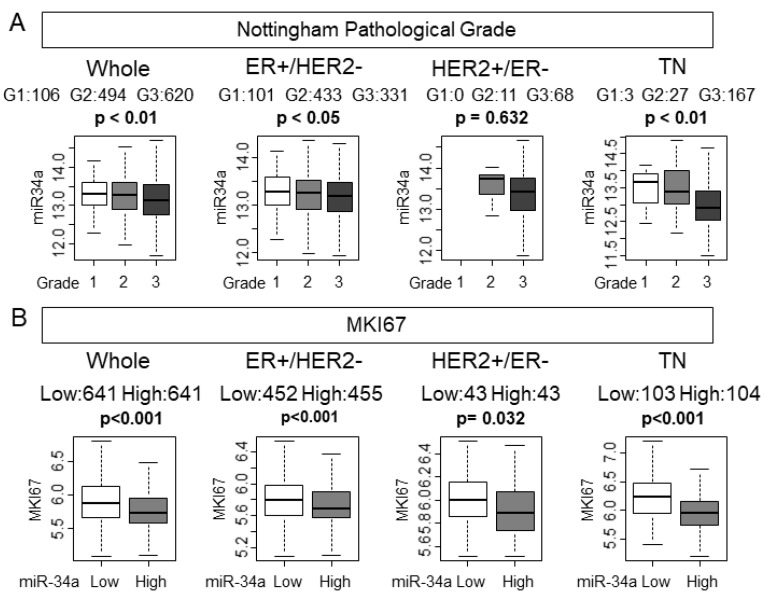
Nottingham pathological grade was associated with miR-34a expression and miR-34a high expression tumors were associated with lower expression of MKI67 in the METABRIC cohort. (**A**) Association between miR-34a expression levels and Nottingham pathological grade in the whole cohort and each subtype. (**B**) Association between expression levels of miR-34a and MKI67 expression in the whole cohort and each subtype.

**Figure 2 ijms-21-03045-f002:**
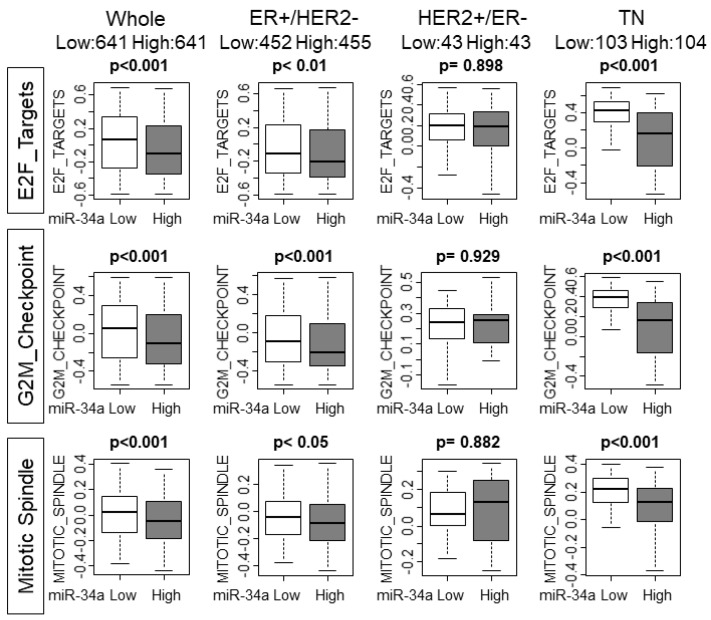
High MiR-34a breast cancer associate with lower gene sets variation analysis (GSVA) scores of the gene sets related to cell cycle or cell proliferation in whole METABRIC cohort and each subtype. Upper row: E2F_Targets, middle row: G2M_Checkpoint, lower row: Mitotic_Spindle Hallmark gene sets. GSVA, Gene Set Variant Analysis.

**Figure 3 ijms-21-03045-f003:**
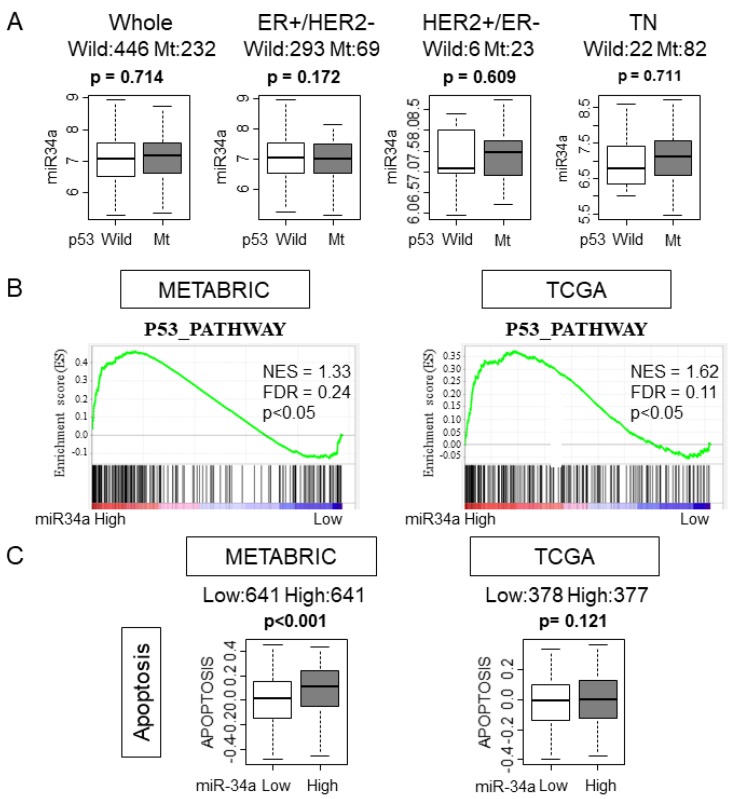
High MiR-34a expression tumors enriched the p53 pathway and apoptosis gene sets. (**A**) Expression levels of miR-34a of p53 wildtype or mutant in the whole cohort and in each subtype in the METABRIC cohort. (**B**) Gene set enrichment analysis (GSEA) of p53 pathway gene sets by miR-34a expression in METABRIC and The Cancer Genome Atlas (TCGA) cohort. (**C**) The association between miR-34a expression and GSVA scoring of Apoptosis in METABRIC and TCGA cohort.

**Figure 4 ijms-21-03045-f004:**
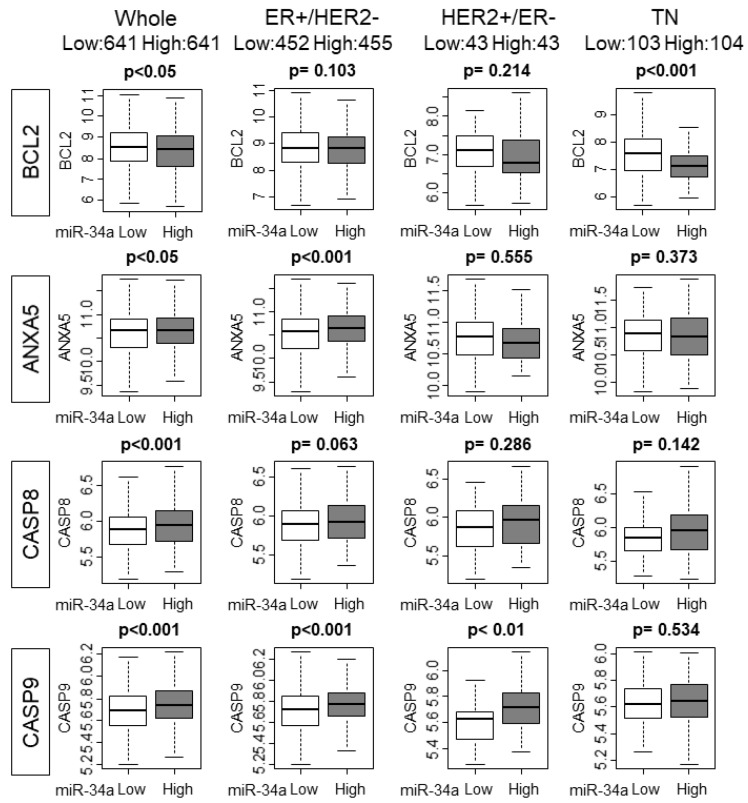
The association of High MiR-34a expression with the expression of apoptosis-related genes in the whole cohort and ER+/HER2−, HER2+/ER−, and triple-negative (TN) subtypes.

**Figure 5 ijms-21-03045-f005:**
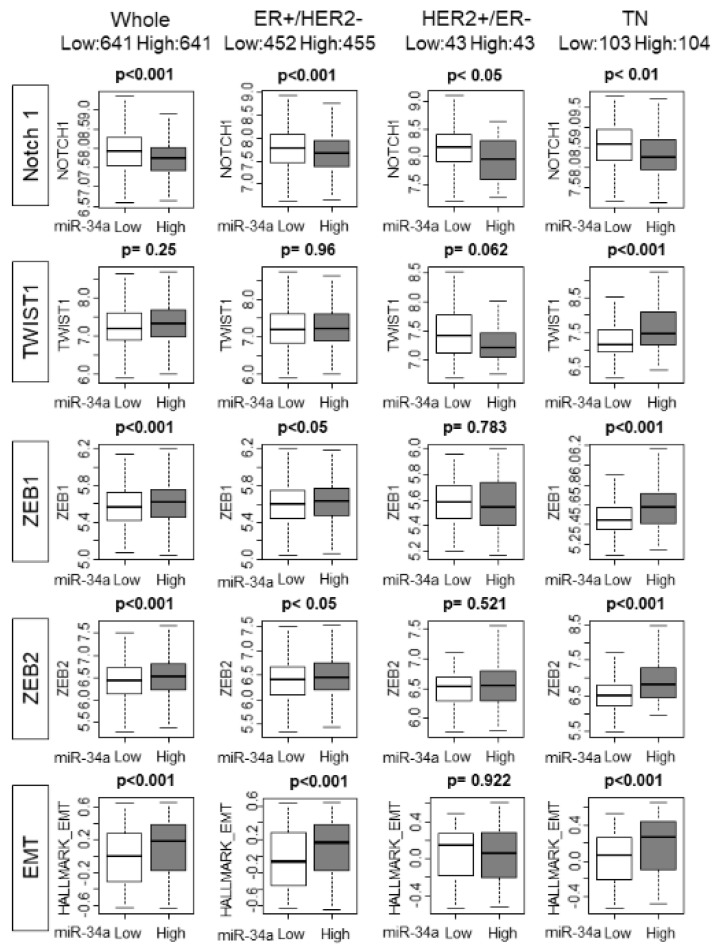
High MiR-34a expression tumors associated with a higher epithelial–mesenchymal transition (EMT) score. Each row has a whole METABRIC cohort, and each subtype. Top row: Notch 1 expression by miR-34a expression. Second row: TWIST 1 expression by miR-34a expression. Third row: ZEB 1 expression by miR-34a expression. Fourth row: ZEB 2 expression by miR-34a. Bottom row: GSVA Hallmark EMT score by miR-34a expression. EMT, epithelial–mesenchymal transition.

**Figure 6 ijms-21-03045-f006:**
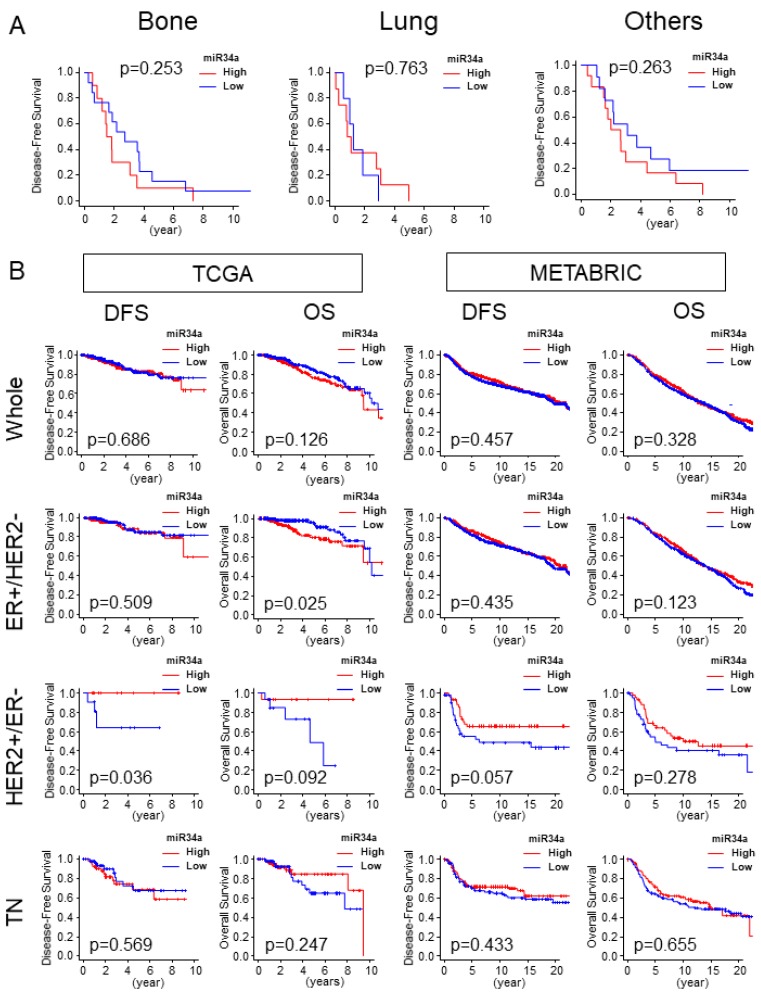
High miR-34a expression tumor was not associated with distant metastasis and was not associated with better survival outcomes. (**A**) Disease-free survival (DFS) of bone, lung, and other metastases in the TCGA cohort. Bone, *n* = 23, high = 10, low = 13; lung, *n* = 13, high = 8, low = 5; *n* = 23, high = 12, low = 11. (**B**) DFS and OS by miR-34a expression in the whole cohort, and each subtype in TCGA and METABRIC. TCGA cohort (whole, *n* = 752, high = 376, low = 376; ER+/HER2−, *n* = 407, high = 204, low = 203; HER2+/ER−, *n* = 30, high = 15 Low = 15; TN, *n* = 117, high = 59, low = 58). METABRIC cohort (Whole, *n* = 1282, High = 641 Low= 641; ER+/HER2−, *n* = 907, High = 455 Low = 452; HER2+/ER−, *n* = 86, High = 43 Low=43; TN, *n* = 207, High = 104 Low = 103). DFS, Disease-Free Survival; OS, Overall Survival.

**Table 1 ijms-21-03045-t001:** Clinicopathological demographics of the high-miR-34a and low-miR-34a groups in the Molecular Taxonomy of Breast Cancer International Consortium (METABRIC) cohort.

Clinicopathological Factor	Whole Cohort (*n* = 1282)	*p* Value
miR-34a High *n* = 641	miR-34a Low *n* = 641
**Age**			
<65 y	259	252	0.732
≥65 y	382	389	
**Stage**			
0	5	6	0.068
1	216	168	
2	296	310	
3	54	46	
4	8	2	
Unknown	62	109	
**Tumor Size**			
≤2 cm	48	47	0.621
2–5 cm	300	314	
>5 cm	285	266	
Unknown	8	14	
**Lymph Node Metastasis**
Negative	319	338	0.954
Positive	276	289	
Unknown	46	14	
**ER Status**			
Negative	141	159	0.262
Positive	500	482	
**PgR Status**			
Negative	292	315	0.218
Positive	349	326	
**HER2 Status**			
Negative	548	573	0.043
Positive	93	68	
**Histologic Subtype**			
Invasive Ductal Carcinoma	481	502	0.254
Invasive Lobular Carcinoma	54	48	
Mixed Ductal and Lobular Carcinoma	75	72	
Other	11	4	
Unknown	20	15

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
