# Peer review of "High Expression of miR-34a Associated with Less Aggressive Cancer Biology but Not with Survival in Breast Cancer"

_ijms, 2020, doi:10.3390/ijms21093045_

Round 1
Reviewer 1 Report
The manuscript of Tokumaru et al investigates the clinical features associated with miR34a expression in breast tumours using public datasets METABRIC and TCGA. Numerous studies investigate the role of miR34a in molecular and cellular processes, and only some studies investigated its clinical relevance. miR34a has been identified as a tumor suppressor gene and the authors claims that high expression of miR34a associate with less aggressive cancer biology with no survival benefit.
Several major points needed to be addressed to improve the current version of the manuscript.
Main comments
- A general major point is that the analyses performed using METABRIC dataset have not been analysed or confirmed using TCGA dataset. Usually, two rounds of analyses should be performed with a training and validation series. All the validation analyses should thus be performed.
- In Table 1, how is obtained the ER/PR/HER2 status? Does it correspond to PAM50 or IHC data? A table summarizing the clinical parameters of the used METABRIC and TCGA cohorts should be provided. Moreover, the number of patients in each category should be given (Figures 1 to 5) as well as the median survival of both TCGA and METABRIC cohorts.
- Figure 1A. Based on the graphs, no significant change in Nottingham Pathological Grade can be seen on the box plot. What statistical test has been performed? Does it correspond to Fisher’s exact test (as for the table)? If yes, not sure these kind of graphs are the best to illustrated the statistical comparison.
- Figure 2. How the authors explain the fact that in HER2+, miR34a expression is significantly associated with Ki67 but not with score of cell progression pathways in GSVA?
- Figure 4. When analysing relationship between EMT and miR34a, do the authors analyse both patients that will develop or not metastasis? Since EMT is link to metastasis process, it could be important to compare this EMT/miR34a association in patients with either metastasis at diagnosis or that have developed metastasis.
- Figure 4. EMT is associated with Claudin-low subtype, which corresponds to breast tumours usually present in all breast cancer subtype. Since only a slight difference is observed between miR34a expression and EMT transcription factor in almost all cancer type, analysing Claudin-low vs others breast cancer would clearly demonstrate the relationship between high miR34a expression and Claudin-low tumors.
Minor comments
- line 38/line72. “by interaction with” should be changed as it can imply a physical interaction between miRNA and p53 while here it corresponds to “by playing a role in”.
- line 65-66. Sentence is difficult to understand.
- Usage of either GSVA and GSEA should be explained in the Mat&Met section.
- Table 1. ER negative/positive. Is “48” correct? (Number corresponding to ER+ in miR34a/low). If yes, there is a statistical problem. “Lympn node metastasis” should be corrected.
- Figure 3D. Difference in expression of all apoptotic genes should be given for all breast cancer subtypes. It has to be noted that the difference in gene expression in high and low miR34a groups is very small.
Reviewer 2 Report
Major concern: This study analyzed existing data obtained from public portals. Only transcriptomic data were included. Major results are associations observed between miRNA34a expression levels and that of other sets of genes. It will be more convincing if the authors could verify some of the gene sets by protein expression analyses, such as by immunoblotting.
Minor: This manuscript need to be further edited by the authors to make font size the same in the main text part.
Round 2
Reviewer 1 Report
The authors have answered all the comments raised. Only few minor points are lacking that will facilitate the reading and appreciation of the data.
Minor comments
- Table 1. Table legend should mention “METABRIC cohort” and the statistical analyses reported from lines 79-82 (SupTable 1) should mention “TCGA cohort”.
- Median follow-up of TCGA and METABRIC should be given in order to interpretate the Figure 6.
Reviewer 2 Report
It is understandable that the authors can not use another method to confirm their results because they do not have access to patients' tissue sample. The authors have added additional discussions.
